# Loss of *dlx5a*/*dlx6a* Locus Alters Non-Canonical Wnt Signaling and Meckel’s Cartilage Morphology

**DOI:** 10.3390/biom13091347

**Published:** 2023-09-05

**Authors:** Emily P. Y. Yu, Vishal Saxena, Sofia Perin, Marc Ekker

**Affiliations:** Department of Biology, University of Ottawa, Marie-Curie Private, Ottawa, ON K1N 94A, Canadasperin@uottawa.ca (S.P.)

**Keywords:** zebrafish, *dlx*, Wnt signaling, craniofacial development, neural crest cells

## Abstract

The *dlx* genes encode transcription factors that establish a proximal–distal polarity within neural crest cells to bestow a regional identity during craniofacial development. The expression regions of *dlx* paralogs are overlapping yet distinct within the zebrafish pharyngeal arches and may also be involved in progressive morphologic changes and organization of chondrocytes of the face. However, how each *dlx* paralog of *dlx1a*, *dlx2a*, *dlx5a* and *dlx6a* affects craniofacial development is still largely unknown. We report here that the average lengths of the Meckel’s, palatoquadrate and ceratohyal cartilages in different *dlx* mutants were altered. Mutants for *dlx5a^−/−^* and *dlx5i6^−/−^*, where the entire *dlx5a/dlx6a* locus was deleted, have the shortest lengths for all three structures at 5 days post fertilization (dpf). This phenotype was also observed in 14 dpf larvae. Loss of *dlx5i6* also resulted in increased proliferation of neural crest cells and expression of chondrogenic markers. Additionally, altered expression and function of non-canonical Wnt signaling were observed in these mutants suggesting a novel interaction between *dlx5i6* locus and non-canonical Wnt pathway regulating ventral cartilage morphogenesis.

## 1. Introduction

The craniofacial skeleton is a complex system of interconnected structures where correct morphology is indispensable for proper function. The cartilage and bones that make up the facial skeleton are derived from cranial neural crest cells (NCCs), a multipotent lineage that migrate from transient segments of neuroepithelium behind the hindbrain, called rhombomeres [1]. Cells from the first two segments then populate transient stripes called pharyngeal arches [2,3,4]. The cranial NCCs from the first pharyngeal arch (PA1) form the Meckel’s cartilage (MC) and the palatoquadrate (PQ), while the second arch forms the ceratohyal (CH), among other structures. Based on their positional identity within rhombomeres, cranial NCCs express different proteins that will determine their final destination [5]. For example, *hox* gene expression provides patterning information along the rostral–caudal axis, whereas *dlx* expression patterns cranial NCCs along the dorsal–ventral axis [6,7,8,9].

Once cranial NCCs have migrated to their final location, they undergo differentiation, becoming chondrocytes that will become the cartilage that supports facial skeletal development [10]. Chondrocytes also undergo convergence and extension movements that allow for a tissue to elongate and narrow in perpendicular axes, which is essential for the proper shape of craniofacial structures [11]. This morphogenic movement is governed by the activity of non-canonical Wnts that can alter, among other processes, oriented cell division and cellular morphology [12]. Indeed, knock-down or knock-out of members of the non-canonical wnt pathways, such as *wnt5b*, *wnt9a*, *frzd7a*, *ror2* and *wls*, resulted in reduced size of PA1 and PA2 structures and changes to chondrocyte cell morphology in zebrafish [11,13,14,15,16,17]. Futhermore, loss of *wnt5b*, *wnt9a* and *ror2* in zebrafish lead to loss in cell polarity in craniofacial structures [11,15]. However, the upstream signals or genes that regulate the non-canonical Wnt pathway during morphogenesis of jaw structures are still poorly understood.

The *dlx* genes encode homeobox-binding transcription factors that are arranged in convergently transcribed pairs [18]. Jawed vertebrates have at least six paralogs, all of which are expressed in overlapping domains within the pharyngeal arches [8,19,20,21]. Despite these overlapping domains, certain *dlx* genes are essential for different parts of facial elements. In the mouse, *Dlx1* and *Dlx2* are important for the development of the maxilla while *Dlx5* and *Dlx6* are essential for the development of the mandibles [21]. To further delineate boundaries of *dlx* activity during craniofacial patterning, zebrafish were used to study morphogenesis of PA1 and PA2 structures, identifying regions of relative importance. The activity of *dlx2a* is important for patterning of dorsal structures and *dlx5a* important for patterning slightly more ventral structures albeit not the most ventral elements [22]. Additionally, facial abnormalities resulting from reduced *dlx1a/dlx2a* function may also be due to increased apoptosis of cranial neural crest cells [23]. However, attempts to elucidate the effect each *dlx* gene is exerting on the development of PA1 and PA2 structures were carried out through gene knock-down or in hypomorphic mutants, and may not be truly representative of gene function [24].

To study the effects of each *dlx1a*, *dlx2a*, *dlx5a* and *dlx6a* during craniofacial development, mutants were generated using a CRISPR-Cas9 mutagenesis approach to delete the entire gene of interest [25]. Additionally, mutants were created where the intergenic enhancers of *dlx5a/dlx6a* were deleted (*inter56*) as well as a line where the entire *dlx5a/dlx6a* locus, including the intergenic region was deleted (*dlx5i6*). Using these mutants, it was found that each *dlx* gene has differential impact on the size of PA1 and PA2 derivatives. Most strikingly, *dlx5i6* mutants presented with shortened and malformed MC, increased proliferation of NCC, changes in the expression of genes important for chondrogenesis and most curiously, alterations in the expression of non-canonical Wnt signal *wnt5b* and its potential receptors *ror2* and *frzd7a*. Within the modified MC, *dlx5i6* mutants also resulted in defects in cell polarity, further supporting a novel interaction between *dlx* and non-canonical Wnt activity.

## 2. Materials and Methods

### 2.1. gRNA Construction and Germline Transformation

All guide RNAs (gRNAs) were designed using CHOPCHOP to target (1) either the 5′UTR or the first exon and (2) the last exon or the 3′UTR of each gene (Appendix A).

To delete the *dlx5a/dlx6a* intergenic region, guides were designed to target the 3′UTR of *dlx6a* and a region 3′ of the *i56ii cis* regulatory element, that is close to *dlx5a*.

To delete the entire *dlx5a/dlx6a* locus, a gRNA targeting the first exon of *dlx6a* and one targeting the first exon of *dlx5a* were designed. All gRNAs were subsequently synthesized as previously described [26].

Cas9 protein and both gRNAs targeting the same gene or desired region were co-injected in 1-cell stage embryos. Primary injected embryos were raised to sexual maturity and crossed to wild-type fish. Resulting F1 embryos were screened using PCR. To detect the deletion of the entire gene or region, two primers were designed to flank the gRNA target sites (F1/R1) and a third primer was designed to target within the gene/region (e.g., R2). If the targeted region was successfully excised, only the product produced by the primers flanking the gRNAs (F1/R1) would appear on the gel. However, if the region was intact or if cutting of only one gRNA occurred, the product produced by F1 and R2 would be visible while the F1/R1 product is too large to be amplified by PCR conditions (Appendix A). Mutant PCR products were also sequenced to confirm loss of targeted region.

The following tables list the sequences used to generate gRNA (Table 1) and the primer sequences for genotyping each mutant (Table 2).

### 2.2. Animal Care and Husbandry

All experiments were conducted following protocols approved by the University of Ottawa Animal Care Committee. Adult zebrafish were housed in circulating water at 28.5 °C with a 14 h light cycle. Embryos were collected from natural spawning and raised in E3 embryo media at 28.5 °C. Larvae for whole mount in situ hybridization, acridine orange staining, or MTOC staining were treated with phenylthiourea (final concentration of 0.003%) at 24–26 hpf to prevent development of pigmentation. Larvae for histology stains at 14 dpf were raised in petri dishes and fed a diet of live rotifers twice daily with water changes from 5 dpf onwards.

### 2.3. Histology Staining

Alcian blue and alizarin red staining in 5 dpf larvae was performed following established protocols [27]. For staining in 14 dpf larvae, the following modifications were used: larvae were fixed in 2% PFA for 1.5 h at room temperature then washed in 10 mM Tris, pH 7.5/10 mM MgCl_2_ for 15 min prior to overnight staining with 10 mM MgCl_2_/0.04% alcian blue. The next day, samples were rehydrated stepwise with 80%, 50% and 25% ethanol/100 mM Tris washes for 15 min each. Samples were then washed with 0.5% KOH for 15 min and then bleached with 3% H_2_O_2_/0.5% KOH for 15 min then rinsed with 35% saturated NaBO_4_ for 15 min. A solution of 1% trypsin in saturated NaBO_4_ was used to clear some larval tissues. Samples were then washed with 10% glycerol/0.5% KOH until saturated before staining overnight with 0.01% alizarin red, pH 7.5. Samples were cleared with 50% glycerol/0.1% KOH until saturated and then an overnight wash before imaging in 70% glycerol/H_2_O.

### 2.4. Whole Mount In Situ Hybridization

Antisense mRNA probes were labelled with digoxygenin-dNTP (Roche, Basel, Switzerland, #11277073910) and synthesized from cDNA clones (*sox9a*, NM161343) [28] or created by PCR using the following primers (Table 3).

The T7 and T3 promoter sequences were appended to the 5′ end of the correct primer for RNA probe synthesis. Whole mount in situ hybridization was performed following an established protocol [29]. Images are representative of at least 15 larvae per time point per target.

### 2.5. BrdU Treatment

At 52hpf (*dlx5i6*^+/−^; *fli1a*:*GFP* incross) or 5 dpf (*dlx5i6*^+/−^; *sox10*:*GFP* incross) larvae were treated with either 10 mM BrdU or 15 mM BrdU, respectively, following established protocols. Larvae were then washed three times with system water and recovered for 4 h at 28.5 °C in system water. At 4 h post treatment, all treated larvae were euthanized with tricaine before decapitation below the yolk sac. Heads were fixed in 2% PFA for 30 min at room temperature in 96 well plates and the posterior portion lysed for genotyping in 10 uL 0.05 mM NaOH. After PFA fixation, samples were washed with PBS and then stored in 100% methanol at −20 °C until genotypes were identified.

### 2.6. Acridine Orange Staining

For acridine orange staining, live embryos at 55 hpf were stained with 10 ug/mL of acridine orange (3, 6-Bis(dimethylamino)acridine hydrochloride zinc chloride double salt; Sigma-Aldrich, St. Louis, MO, USA) in system water for 30 min in the dark at room temperature (~23 °C). Larvae were washed three times with system water prior to imaging.

### 2.7. MTOC Sample Preparation and Immunohistochemistry

For BrdU staining, samples were rehydrated step-wise in 75%/50%/25% methanol/PBST then washed in PBST before permeabilized with 10 mg/mL proteinase K for 20 min (56 hpf) or 40 min (5 dpf). Samples were then post-fixed with 4% PFA for 20 min followed by 2 N HCl for 30 min (56 hpf) or 1 h (5 dpf) for antigen retrieval. All samples were blocked with 10% FBS/PBST for 1 h.

To stain the microtubule organizing center (MTOC), larvae were euthanized in excessive tricaine and then fixed in 4% PFA overnight at 4 °C. Samples were washed with PBST then cut below the yolk ball for greater antibody penetration and mounting for imaging. Samples were then digested with 10 mg/mL proteinase K for 30 min and then washed 3 times 15 min with 0.1% PBS-TritonX (PBSTr) before blocking with 10% FBS/0.5%PBSTr for 2 h.

Staining for Sox10 was performed following previously established protocols using PBST [30].

The following primary antibodies and concentrations were used: 1:200 mouse anti-acetylated tubulin (Sigma, T7451), 1:400 mouse anti-BrdU (Sigma, B2531), 1:500 rabbit anti-GFP (Life Technologies, Carlsbad, CA, USA, #A11122). The following secondary antibodies were used, all at 1:1000 concentration: goat anti-mouse Alexa 488 (Life Technologies, #A11001), goat anti-mouse Alexa 594 (Life Technologies, #A11005) and goat anti-rabbit Alexa 488 (Life Technologies, #A11008).

Alexa-Fluor-594 phalloidin (Thermo Fisher Scientific, Waltham, MA, USA, #A12381) was used at 3:100 during secondary antibody incubation.

### 2.8. RNA Extraction, cDNA Synthesis and RT-qPCR

Total head RNA was extracted from 2 dpf and 3 dpf *dlx5a*^−/−^ and WT sibling larvae with TriZol (Invitrogen, Waltham, MA, USA, Thermo Fisher, Waltham, MA, USA) according to manufacturer’s protocol. A pool of 20–25 embryos were used for each timepoint from different crosses to generate at least four biological samples. After precipitation, all RNA were analyzed for purity and integrity through gel electrophoresis and using the NanoDrop 1000 spectrophotomerter (Thermo Fisher Scientific). Samples with absorbance of 1.95–2.1 and clear 18S and 28S bands were used for cDNA synthesis. For cDNA synthesis, 750 ng of RNA was used for iScriptTM cDNA Synthesis Kit (Life Science Research, Bio-Rad, Hercules, CA, USA) following the manufacturer’s protocol. RT-qPCR was performed in triplicate with 1:4 cDNA diluted in nuclease-free water and using SsoFastTM EvaGreen^®^ Supermix (Bio-Rad) on the Bio-Rad CFX96 instrument. The following table lists primers used for *sox9a*, *col11a2*, *wnt5b*, *ror2* and *frzd7a* (Table 4). For reference genes, *ef1a* [31] and *r18s* [32] were used. Primer efficiencies for all targets were between 90–113%. Analysis of fold changes was performed using Microsoft Excel 2021.

### 2.9. Imaging

Whole mount in situ hybridization images were acquired using Nikon SMZ 1500 and NIS-Elements F software. BrdU images were acquired using the Zeiss LSM 880 confocal with the 10× objective. Acridine orange images were obtained using the Zeiss AxioZoom V16 with Apotome. To image all other immunohistochemistry experiments, the Nikon AIRsIMP confocal was used with 20× water immersion objective. For all confocal imaging, samples were mounted in 1% low-melt agarose. Whole mount in situ samples were imaged in 70% glycerol/PBS on a glass slide or cover slip. For acridine orange images, live larvae were mounted in 3% methylcellulose made with system water.

### 2.10. Cell Counting and Statistical Analysis

All cell counting was performed on z-stacks using ImageJ software (version 1.53o) with the BioFormats plug-in [35,36]. Graphs and statistical analysis were performed using Graphpad Prism 9 (San Diego, CA, USA). For average cartilage lengths, one-way ANOVA with Tukey’s multiple comparisons test was used. All other statistics used two-tailed unpaired *t*-test with Welch correction and Holm Sidak method. Statistical significance was determined with a 95% confidence interval where * *p* < 0.05; ** *p* < 0.001; *** *p* < 0.0001.

## 3. Results

### 3.1. Loss of Function Mutants Are Viable and Do Not Exhibit Severe Morphological Defects

We generated a collection of *dlx* mutants using CRISPR-Cas9. The approach involved using pairs of guide RNAs positioned in such a way that most (>75%; *dlx2a*, *dlx5a*, *dlx6a*) or all (*dlx1a*) of the coding region of the gene(s) was deleted. All deletions included the homeobox. For the *dlx5i6* mutants, the two guide RNAs were positioned such that >75% of the coding region of the *dlx5a* and *dlx6a*, as well as the intergenic region separating the two genes was deleted (Appendix A). The expected deletions were verified by PCR and sequencing (Appendix A).

We previously reported all *dlx* mutants were viable and fertile as adults [25]. Surprisingly, all mutants did not have severe or consistent morphological craniofacial defects as adults. All fish are able to feed on a standard diet of rotifers, brine shrimp and flake food.

### 3.2. Morphology of Jaw Structures Are Altered in dlx Mutants at 5 dpf and 14 dpf

To observe if defects occurred during craniofacial development in *dlx* mutants, alcian blue and alizarin red staining were performed on 5 dpf larvae (Figure 1A–G). Flat mounted images in the ventral position for *dlx5a^−/−^*, *dlx5i6^−/−^* and WT siblings were also produced to better showcase abnormal morphology of key jaw structures (Figure 1H–K). The average lengths of the MC, PQ and CH were obtained by measuring each half of each structure of an animal, averaging the measurement then averaging for each genotype (Figure 1L). The MC was found to be shorter in *dlx5a^−/−^* and *dlx5i6^−/−^* mutants compared to WT (Figure 1M). The PQ was shorter in *dlx1a^−/−^*, *dlx2a^−/−^*, *dlx5a^−/−^* and *dlx5i6^−/−^* mutants compared to WT, (Figure 1N). Other than *dlx5a^−/−^* and *dlx5i6^−/−^* larvae, *dlx2a^−/−^* mutants also possessed shorter CH (Figure 1O). Interestingly, the overall length of the head was shorter in *dlx1a^−/−^*, *dlx2a^−/−^*, *dlx5a^−/−^* and *dlx5i6^−/−^* mutants, suggesting an early growth defect (Appendix A). Importantly, the expression of *dlx2b*, a duplicate of *dlx2a*, was similar to WT siblings in *dlx2a^−/−^* larvae at 2 dpf and 3 dpf, suggesting craniofacial defects observed in *dlx2a^−/−^* larvae are likely not due to *dlx2b* compensation (Appendix A).

Since no overt morphological defects were observed in adults, it was possible that defects observed at 5 dpf may be rescued or normalized at a later age. Thus, morphometrics of the same three cartilage structures were made at 14 dpf using the same histology stains in *dlx* mutants (Figure 2A–G). Flat mounted images in *dlx5i6^−/−^* and WT siblings were also taken in ventral position (Figure 2H,I). The MC continued to be shorter in *dlx5a^−/−^* and *dlx5i6^−/−^* mutants compared to WT (Figure 2J). The structure in *dlx1a^−/−^* larvae are now also shorter but only when compared to WT. At this age, the PQ of *dlx1a^−/−^*, *dlx5i6^−/−^* and *inter56^−/−^* mutants were shorter compared to WT and *dlx2a^−/−^* larvae (Figure 2K). Surprisingly, only *dlx1a^−/−^* larvae had shorter CH structures compared to WT, but all other mutants possessed shorter CH when compared to *dlx2a^−/−^* larvae (Figure 2L). Taking the measurements made at 5 dpf into consideration, *dlx5a^−/−^* and *dlx5i6^−/−^* mutants consistently possess shorter MC and PQ while the CH length may have normalized. In the case of *dlx1a^−/−^* mutants, the three cartilages are now shorter at 14 dpf which may reflect the fact the head length of these mutants continue to remain shorter than WT (Appendix A).

The ratio of how far distally the arch of the MC was extended (length) and the distance between the two arms of the structure (width) was calculated. At both ages, *dlx5a^−/−^* and *dlx5i6^−/−^* mutants possess smaller ratios (Figure 1P and Figure 2M). In sum, *dlx5a^−/−^* and *dlx5i6^−/−^* mutants not only possessed a shorter MC, but the structure was wider compared to WT and other mutants. Since *dlx5i6^−/−^* larvae possessed more severe defects at 5 dpf, these mutants were the subject of further analyses.

### 3.3. Altered Expression of Chondrocyte Markers and Increased Proliferation Observed in dlx5i6 Mutants

A possible explanation for why *dlx5i6^−/−^* possessed shorter and malformed PA 1 structures may be that fewer NCC contribute to these structures, either through altered migration or differentiation. To test this, the expression of *foxd3*, *sox9a* and *col11a2* was assessed by whole mount in situ hybridization (WISH).

At 9 hpf, cells at the neural plate border express *foxd3* as they undergo induction. In *dlx5i6^−/−^* embryos, staining for *foxd3* revealed the neural plate border is further apart than age-matched WT, suggesting possible defects in NCC positioning or migration as the embryo develops (Figure 3A,B). However, by 14 hpf, after the onset of NCC migration, the pattern of *foxd3* staining is similar in *dlx5i6*^−/−^ and WT, suggesting overall migration was not affected (Figure 3C,D). To further explore NCC migration, immunohistochemistry was performed at 16 hpf for Sox10 *dlx5i6^−/−^* embryos (Figure 3E,F). No significant differences were observed in the number of Sox10^+^ cells that have migrated from the neural tube at 16 hpf (Figure 3G). These results together further suggest migration is likely unaffected in *dlx5i6* mutants.

Could altered craniofacial structure size be due to insufficient cell proliferation once NCC were in the pharyngeal arches? To determine if cell proliferation was affected in these mutants, embryos were treated with BrdU at either 50 hpf or 5 dpf with processing carried out 6 h later. At both timepoints, more BrdU+ cells were present in *dlx5i6^−/−^* larvae (Figure 4D,F) compared to WT siblings (Figure 4C,E). Many BrdU+ cells were also positive for GFP controlled by either the *fli1a* or the *sox10* promoter, suggesting increased proliferation was occurring in NCC derivatives. At both timepoints, there was a near two-fold increase in the number of BrdU+ cells that co-expressed GFP in mutants compared to WT (Figure 4G). Importantly, increased apoptosis was not observed in *dlx5i6^−/−^* at 55hpf demonstrating the change in proliferation was not to offset cell death previously reported in knock down of *dlx* genes in zebrafish larvae (Figure 4A,B).

To probe NCC differentiation, the expression of the transcription factor *sox9a*, which is expressed during specification and is essential for the differentiation of NCC into chondrocytes was investigated [37,38,39]. At 2 dpf, expression *sox9a* was observed in the first and second pharyngeal arches in both WT and *dlx5i6*^−/−^ larvae, further supporting the possibility that NCC migration was unaffected (Figure 5A,B). However, by 3 dpf, the expression pattern of *sox9a* differs in *dlx5i6^−/−^* compared to age-matched WT, where staining was observed throughout ventral cartilaginous structures (Figure 5C,D). The expression of *col11a2*, a minor collagen present in facial cartilage was also investigated [40]. At 2 dpf, both WT and *dlx5i6^−/−^* larvae have comparable expression pattern and staining intensity (Figure 5E,F). However, by 3 dpf, the staining for *col11a2* in *dlx5i6^−/−^* larvae appear more pronounced in the ethmoid plate and MC (Figure 5G′,H′). Since WT and mutant siblings were treated and stained for the same amount of time for each marker, the changes in staining pattern and intensity at 3 dpf suggest *dlx5i6*^−/−^ larvae may exhibit increased expression of *sox9a* and *col11a2*. Based on results from BrdU experiments, this may also be a consequence of more cells contributing to facial structures. Further investigation was performed by quantifying gene expression through reverse-transcription quantitative PCR (RT-qPCR) on cDNA of pooled larval heads at 2 dpf and 3 dpf, for *sox9a* and *col11a2*. Expression of *sox9a* was similar in both WT and mutants at 2 dpf but showed a trend towards increased expression at 3 dpf (Figure 5I). The expression of *col11a2* showed a trend towards higher expression in mutants at both timepoints (Figure 5J).

### 3.4. Non-Canonical Wnt Signaling in MC Is Abnormal in dlx5i6^−/−^

The MC of *dlx5i6^−/−^* larvae is shorter compared to WT; however, NCC contribution to the structure and differentiation into chondrocytes appear unaffected in these mutants. Moreover, increased proliferation in NCC and derivatives were observed suggesting more cells are present to contribute to MC formation. To address these two conflicting lines of evidence, another aspect of MC morphology was considered. The MC in *dlx5i6^−/−^* is wider and fails to extend distally, which resembles mutants of non-canonical Wnt signaling such as *wnt5b*, suggesting non-canonical Wnt signaling may be affected in *dlx5i6^−/−^* larvae [11,17].

Using WISH and RT-qPCR, the expression of *wls*, which controls release of Wnt5b into the extracellular space, and *wnt5b* were investigated (Figure 6). At 2 dpf, loss of *dlx5i6* leads to similar staining of *wls* (Figure 6A,B). However, by 3 dpf, the staining in *dlx5i6*^−/−^ larvae is darker, suggesting increased expression since both biological groups were stained for the same amount of time (Figure 6C,D). In the case of *wnt5b*, 2 dpf mutant larvae exhibited reduced staining compared to WT, suggesting reduced expression (Figure 6E,F). However, by 3 dpf, the staining for *wnt5b* is darker in mutants compared to WT (Figure 6G,H). The expression of *wnt5b* was also investigated by RT-qPCR, confirming reduced expression at 2 dpf in *dlx5i6* mutants (Figure 6I). However, only a trend towards increased expression was observed at 3 dpf. Two receptors for *wnt5b*, Frizzled7a and Ror2 were also investigated by RT-qPCR. Frizzled7a (*frzd7a*) is a potential receptor for Wnt5b and was shown previously to be important in lower jaw development in zebrafish [41,42]. The expression of *frzd7a* show a trend towards higher expression in mutants at 2 dpf followed by lower expression at 3 dpf, potentially to compensate for different amounts of Wnt5b (Figure 6J). The tyrosine kinase receptor Ror2 can interact with Wnt5b during gastrulation in zebrafish and a *ror2* null mutant was recently reported to produce craniofacial abnormalities including altered chondrocyte polarity [15,34,43]. Interestingly, the expression of *ror2* showed a trend towards more expression at both time points in mutants (Figure 6K). Taken together, these analyses suggest that non-canonical Wnt signaling is mis-expressed in *dlx5i6^−/−^* larvae during critical timepoints of MC morphogenesis.

To investigate if altered expression of non-canonical Wnt signaling functionally impacted the morphology of the MC, alcian blue histology stains were revisited to determine whether convergence and extension was affected in the MC. The length and width of a single half of the MC was measured, finding that the L/W ratio was smaller in *dlx5i6^−/−^* larvae at both 5 dpf and 14 dpf (Figure 7A–C), suggesting that the cartilage itself is wider in mutants. Cell polarity was also investigated in *dlx5i6* null mutants through immunohistochemistry for the microtubule organizing center (MTOC) in mutant MC chondrocytes (Figure 7D,E). The MTOC sits at the base of the primary cilium present in most cells and the positioning of the MTOC is regulated by non-canonical Wnt signaling [44]. It was previously reported that the MTOC in chondrocytes to the left of the midline should be skewed towards the left side of the cell while MTOC in chondrocytes to the right of the midline are skewed to the right side of the cell [11]. This assay revealed the positioning of the MTOC in *dlx5i6^−/−^* was more randomized compared to WT siblings (Figure 7F). Additionally, the chondrocytes of *dlx5i6^−/−^* larvae failed to achieve the standard elongated shape observed in WT which is also regulated by non-canonical Wnt signaling [45]. These data suggest loss of Dlx5a/Dlx6a function may impair expression and function of non-canonical Wnt signaling, resulting in morphological defects to the MC.

## 4. Discussion

The *dlx* genes are involved in providing NCC with positional information that will influence their migration from the neural tube. These genes are also involved in providing regional identity during morphogenesis of facial structures. However, the role of each *dlx* gene during craniofacial development is still largely unknown. Here, we report the consequence of *dlx1a*, *dlx2a*, *dlx5a* and *dlx6a* loss of function as well as compound mutants on the development of ventral structures of the first and second pharyngeal arches in zebrafish.

### 4.1. Reduced Length of Meckel’s, Palatoquadrate and Ceratohyal Cartilages in dlx Mutants

At 5 dpf, the MC, PQ and CH of *dlx5a* and *dlx5i6* mutants are all shorter compared to WT. However, by 14 dpf, only the MC is shorter in both mutants. These results suggest *dlx5a* may be more important for regulating the morphology of these structures than other *dlx* paralogs. These results also support previous work using zebrafish that identified *dlx5a* is involved in delineating the intermediate region of ventral cartilaginous structures including the entire MC [22]. However, our work may suggest additional paralogs such as *dlx3b* and *dlx4b* may compensate for *dlx5a* and *dlx6a* loss of function, since loss of both those genes did not result in homeotic transformation of the lower jaw, contrary to observations made in mouse studies [19,20,21,46,47]. The expression of these other paralogs will need to be explored in addition to investigating the morphometrics of these structures in compound mutants. Furthermore *dlx5i6^−/−^* larvae presented with more severe defects compared to loss of *dlx5a* or other *dlx* paralogs. While it was previously demonstrated that loss of *dlx5a* results in loss of *dlx6a* expression, which is also shown in WISH for these genes in *dlx5i6* mutants, it is possible the loss of the entire locus, including the intergenic regulatory elements have affected expression of other genes [25]. It is therefore possible that these unknown genes may also affect MC morphogenesis independent of Dlx5a/Dlx6a activity and warrants further investigation.

Curiously, the structures analyzed in this study were also shorter in *dlx1a* mutants at 14 dpf but not at 5 dpf. At 14 dpf, these pharyngeal skeletal elements would be longer, and the dentary covers the circumference of the Meckel’s cartilage while the other structures are undergoing osteogenesis [48]. Importantly, growth of these elements, particularly the Meckel’s cartilage, is due to division of resident chondrocytes [49]. It is possible that the reduction in the size of these structures in *dlx1a^−/−^* larvae is due to cell division defects in chondrocytes. Additionally, the overall length of *dlx1a^−/−^* larval heads was shorter compared to WT, suggesting a possible general growth defect preventing *dlx1a^−/−^* larval jaw structures from reaching WT lengths.

### 4.2. Increased Proliferation Observed in dlx5i6^−/−^ during Early Pharyngeal Skeletal Morphogenesis

The increased number of BrdU+ cells and the higher expression of markers for NCC and chondrocytes that we observed in *dlx5i6^−/−^* suggest loss of *dlx5i6* locus results in abnormal cell proliferation. This supports previous work in cell lines that showed DLX5 and DLX6 play a role in cell division by antagonizing the G1/S phase transition [50]. It is possible that loss of *dlx5a* and/or *dlx6a* led to increased proliferation through shorter cell cycles. Since NCC proliferation is tightly regulated based on the number of cells in a migratory stream, it is possible that increased proliferation observed in *dlx5i6*^−/−^ may also be due to fewer NCC migrating out from the neural tube [51].

### 4.3. Novel Interaction between dlx5i6 Locus and Non-Canonical Wnt Signaling during MC Morphogenesis

Led by the observation that MC of *dlx5i6^−/−^* larvae phenocopied non-canonical Wnt mutants, particularly of *wnt5b* and *wls*, we investigated expression of Wnt-related genes in *dlx5i6^−/−^* mutants. We observed lower expression *of wnt5b* at 2 dpf and a trend towards increased expression at 3 dpf. The differences in expression of non-canonical Wnt components did not reach significance, potentially because the entire head was used for RNA extraction; expression from additional tissues may be obscuring expression changes only in facial tissue. Future work will require cell sorting to further investigate expression differences in *dlx* mutants.

Both lower and higher expression of non-canonical Wnt signals can lead to cell polarity defects. The positioning of the MTOC in MC chondrocytes was explored finding mutants had a more randomized positioning. Additionally, chondrocytes did not form the expected elongated shape, maintaining a rounded appearance instead. Furthermore, the cartilage itself was also wider at 5 dpf and 14 dpf in *dlx5i6* null mutants, suggesting failures in convergence and extension during MC morphogenesis [12]. These results support a potential link between Dlx function and non-canonical Wnt which was previously unknown in the context of craniofacial development. Whether Dlx5a/Dlx6a is directly upstream of *wnt5b* expression and how the expression changes after 3 dpf requires further investigation. Importantly, Wnt5b is involved in a planar cell polarity pathway to activate effectors JNK and C-jun. However, Wnt5b can also activate an alternative Wnt/Ca^2+^ pathway where NFAT is an effector [52]. Both these pathways alter cell polarity and migration; further analysis would be required to determine whether these pathways are active simultaneously or function independently in craniofacial development and how *dlx* may affect which pathway is active. Additionally, since Wnt5b can also regulate other signaling molecules such as *fgf3* to exert changes in chondrogenic cell proliferation, it is possible these molecules are also mis-expressed in the absence of *dlx5a* and *dlx6a* [17]. The possibility that *dlx* genes can exert local changes to signaling activity in regional and temporal manner may provide answers to how NCC that express similar genetic networks can form such different facial structures.

## 5. Conclusions

The deletion of *dlx* paralogs in zebrafish led to changes in the lengths of cartilage structures of PA1 and PA2 cartilages at 5 dpf and 14 dpf, with the lengths of MC consistently and persistently shorter in *dlx5i6^−/−^*. These mutants also presented with wider MC and increased chondrocyte proliferation. Consistent with these observations, altered expression of the non-canonical Wnt *wnt5b* and other components of the pathway was present in *dlx5i6* mutants. Furthermore, randomized cell polarity, abnormal chondrocyte cell shape and wider cartilage were observed in the MC of *dlx5i6* null animals, suggesting this *dlx* paralog may interact in the non-canonical *wnt5b* pathway.

## Figures and Tables

**Figure 1 biomolecules-13-01347-f001:**
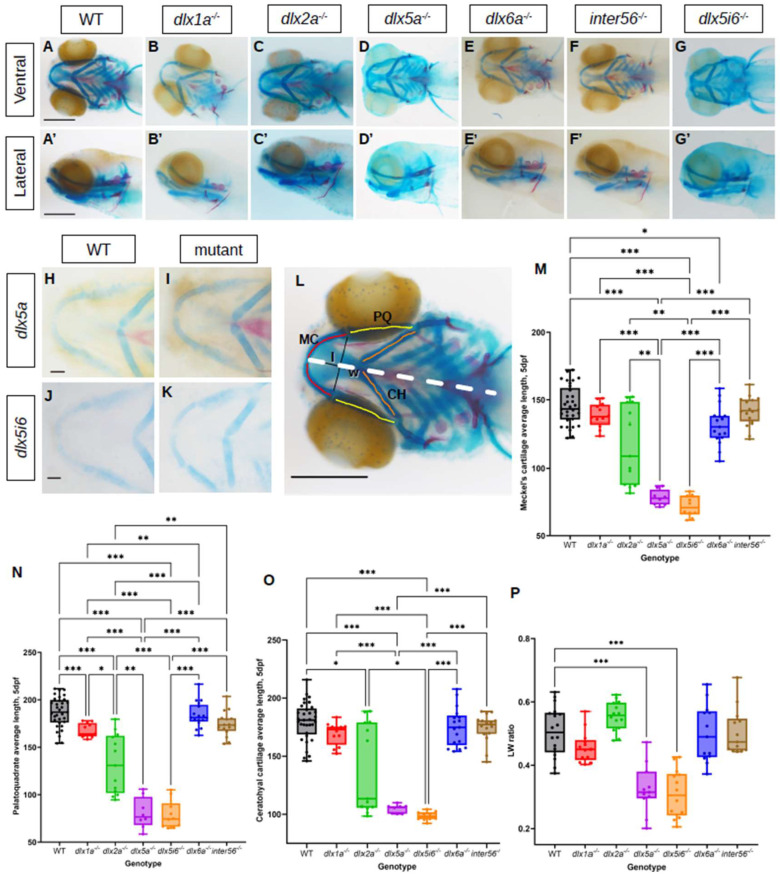
Histological analysis of *dlx* mutants using alcian blue and alizarin red showing altered lengths of MC, PQ and CH cartilage structures at 5 dpf. (**A**–**G**) Histology stains were performed in at least 15 larvae for each genotype and presented in the ventral (**A**–**G**) or lateral (**A′**–**G′**) Scale = 150 μm. (**H**–**K**) Flat mount images in ventral position of *dlx5a^−/−^* (*n* = 5) and *dlx5i6^−/−^* (*n* = 5). Scale = 100 μm. (**L**) Schematic showing how structures were measured. Lengths of both halves of MC (red), PQ (yellow) and CH (orange) were measured, then averaged for each animal. Averages were then calculated per genotype. The length (l) and width (w) of the MC were also measured. Dashed line represents entire head length. (**M**–**O**) Average lengths of MC (**M**), PQ (**N**) and CH (**O**) in WT (*n* = 33), *dlx1a^−/−^* (*n* = 12), *dlx2a*^−/−^ (*n* = 14), *dlx5a^−/−^* (*n* = 8), *dlx5i6*^−/−^ (*n* = 13), *dlx6a^−/−^* (*n* = 17) and *inter56^−/−^* (*n* = 16) larvae. Results are reported as mean ± SEM. (**P**) *l*/*w* ratio in *dlx5a^−/−^* and *dlx5i6^−/−^* larvae were smaller compared to WT. One-way ANOVA with Tukey’s multiple comparisons test was performed. * *p* < 0.05; ** *p* < 0.001; *** *p* < 0.0001.

**Figure 2 biomolecules-13-01347-f002:**
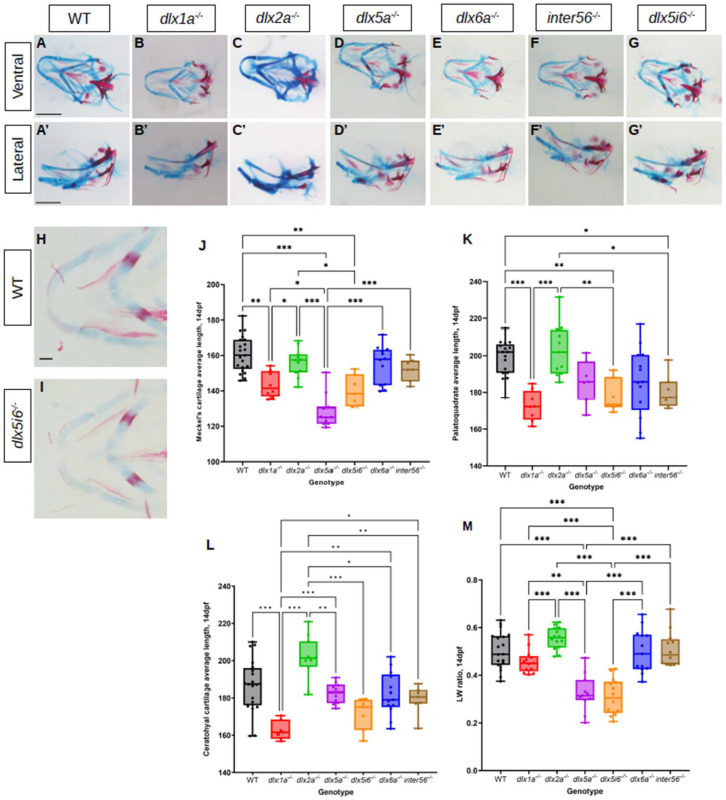
Alcian blue and alizarin red staining of 14 dpf *dlx* mutants reveal altered lengths of craniofacial structures. (**A**–**G**) Histology stains were performed in at least 10 larvae for each genotype. Images were taken in ventral (**A**–**G**) and lateral (**A′**–**G′**) positions. Scale = 150 μm. (**H**,**I**) Flat mount of *dlx5i6^−/−^* (*n* = 5) and WT siblings (*n* = 6) in the ventral position to better show MC structure. Scale = 100 μm. (**J**–**L**) Average lengths of MC (**J**), PQ (**K**) and CH (**L**) in 14 dpf animals were obtained following the same method as 5 dpf. At least 8 animals per genotype were measured. Results are reported as mean ± SEM. (**M**) *l*/*w* ratio in *dlx5a*^−/−^ and *dlx5i6*^−/−^ larvae continue to be smaller compared to WT. One-way ANOVA with Tukey’s multiple comparisons test was performed. * *p* < 0.05; ** *p* < 0.001; *** *p* < 0.0001.

**Figure 3 biomolecules-13-01347-f003:**
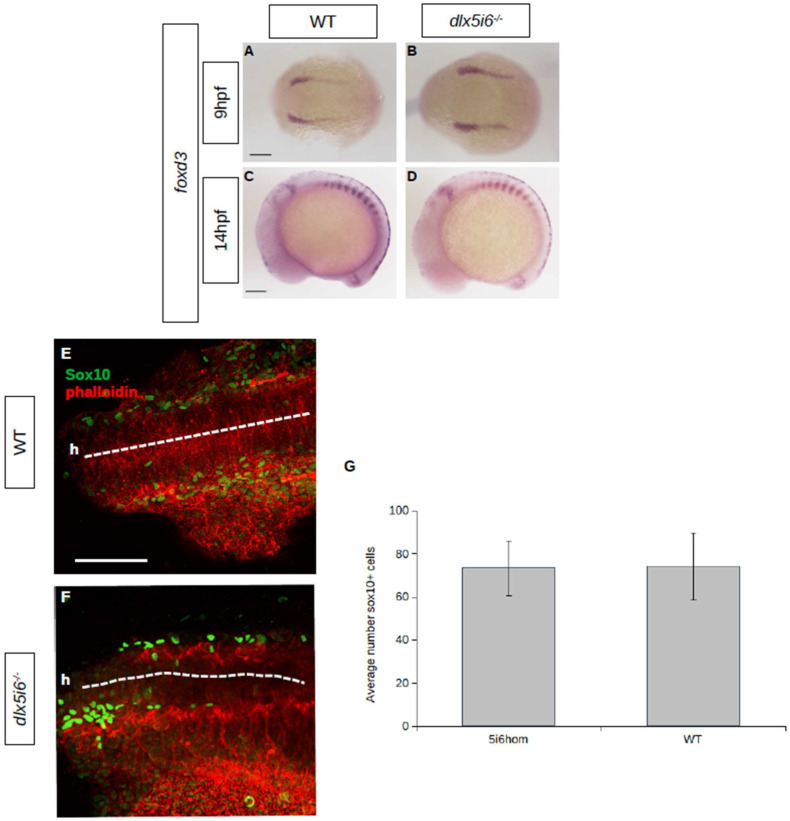
NCC specification and migration does not appear to be affected in *dlx5i6^−/−^* embryos. (**A**–**D**) Expression of NCC specifier *foxd3* by WISH at 9 hpf (**A**,**B**) during NCC specification and at 14 hpf, during NCC migration (**C**,**D**). (**E**,**F**) Immunohistochemistry for Sox10 (green) at 16 hpf in WT (*n* = 5, (**E**)) and *dlx5i6^−/−^* (*n* = 4, (**F**)) show similar number of cells leaving the neural tube (dashed line). ‘h’ represents the location of head. (**G**) Average number of Sox10^+^ cells from second somite to head in a single side of each larvae. Scale = 100 μm.

**Figure 4 biomolecules-13-01347-f004:**
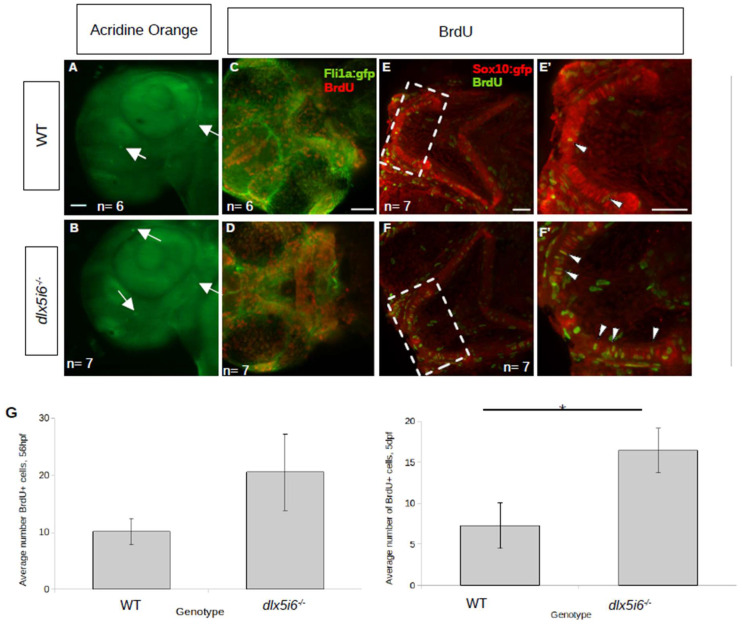
Proliferation is increased in *dlx5i6* mutants at 56 hpf and 5 dpf without changes in cell death. (**A**,**B**) Larvae were stained with acridine orange to label apoptotic cells (arrows) at 55 hpf. WT (*n* = 6, (**A**)) larvae had few labelled cells as was the case in *dlx5i6^−/−^* larvae (*n* = 7, (**B**)) in the pharyngeal arches. Mutants had more labelled cells in other parts of the head. (**C**–**F**) BrdU staining in 56 hpf (**C**,**D**) and 5 dpf (**E**,**F**) larvae. Mutants (*n* = 7 for each timepoint; (**D**,**F**)) had overall more BrdU labelled cells compared to WT siblings (*n* = 6 at 56 hpf, *n* = 7 at 5 dpf; (**C**,**E**)). (**E′**,**F′**) 2× zoom of 5 dpf larvae in area within dashed line box. Arrowheads indicate double-labelled cells. (**G**) Quantification of BrdU+ cells in the MC at 56 hpf (**left**) and at 5 dpf (**right**). * *p* < 0.05, unpaired *t*-test. Scale = 100 μm.

**Figure 5 biomolecules-13-01347-f005:**
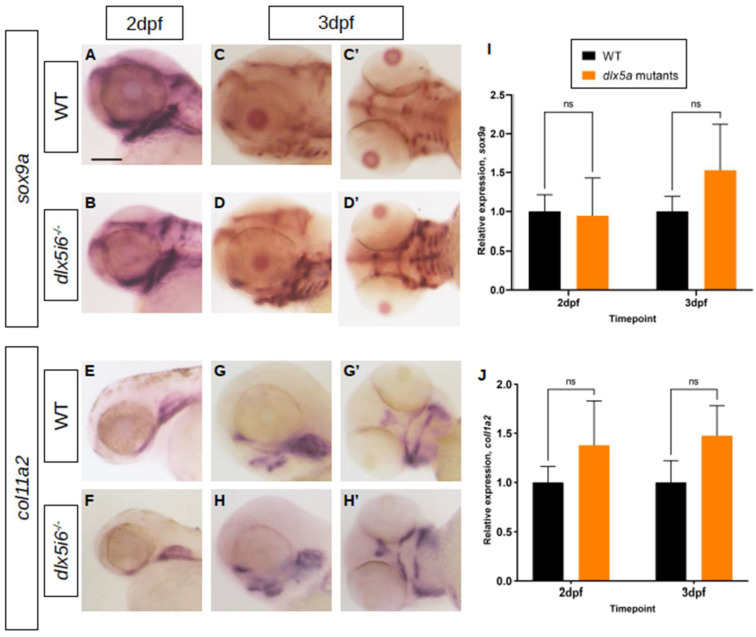
Expression of chondrocyte markers at 3 dpf in *dlx5i6^−/−^* larvae. (**A**–**D**) WISH for *sox9a* at 2 dpf (**A**,**B**) in WT and *dlx5i6^−/−^* larvae in the lateral position. (**C**,**D**) WISH for *sox9a* at 3 dpf in WT and *dlx5i6^−/−^* larvae in lateral (**C**,**D**) and ventral (**C**′,**D**′) positions, showing more staining for this marker in *dlx5i6^−/−^* mutants. (**E**,**F**) WISH for *col11a2* in 2 dpf WT (**E**) and *dlx5i6^−/−^* larvae (**F**) in lateral position. (**G**,**H**) WISH for *col11a2* in 3 dpf WT (**G**,**G**′) and mutant (**H**,**H**′) in lateral and ventral positions. Scale = 100 μm. (**I**,**J**) Relative normalized expression of *sox9a* (**I**) and *col11a2* (**J**) at 2 dpf (*p* = 0.32; *p* = 0.16, respectively) and 3 dpf (*p* = 0.42; *p* = 0.24, respectively). ‘ns’: not significant.

**Figure 6 biomolecules-13-01347-f006:**
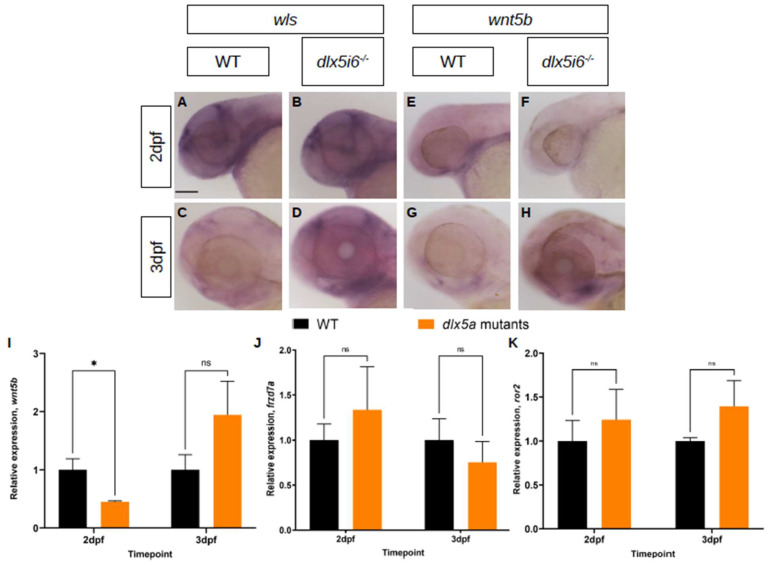
Expression of non-canonical Wnt signaling components are altered in *dlx5i6^−/−^* larvae during MC morphogenesis. (**A–D**) Expression of *wls* at 2 dpf (**A**,**B**) and 3 dpf (**C**,**D**) by WISH. (**E–H**) Expression of *wnt5b* at 2 dpf (**E**,**F**) and 3 dpf (**G**,**H**) by WISH. Scale = 100 μm. (**I–K**) Normalized relative expression of *wnt5b* (**I**), *frzd7a* (**J**) and *ror2* (**K**) by RT-qPCR using at least 4 pools of head cDNA. All data presented as mean ± SEM. * *p* < 0.05, ns = no significance (*wnt5b* at 3 dpf: *p =* 0.21; *frzd7a* at 2 dpf and 3 dpf: *p* = 0.07, *p* = 0.28, respectively; *ror2* at 2 dpf and 3 dpf: *p* = 0.13, *p* = 0.08, respectively).

**Figure 7 biomolecules-13-01347-f007:**
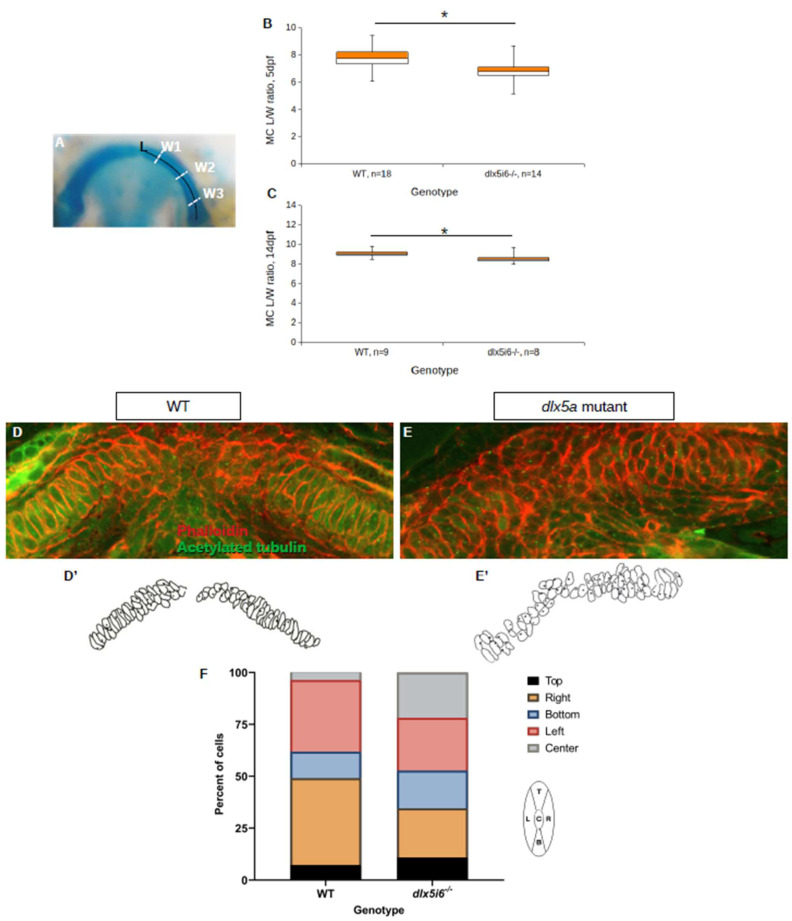
Decreased LW ratio of MC and mispositioning of microtubule organizing center (MTOC) in *dlx5i6^−/−^* larvae. (**A**–**C**) Histology stains were revisited to calculate the length and width of MC in 5 dpf (**B**) and 14 dpf (**C**) *dlx5i6^−/−^* larvae as an indicator of abnormal planar cell polarity. The length of half the MC was measured from the proximal to distal tip (**A**). Three width measurements were made and averaged to account for potential differences along the arch. (**D**,**E**) Larvae at 3 dpf were stained with phalloidin (red) and acetylated tubulin (green) to label cell membranes and MTOC, respectively, in WT (**D**) and *dlx5i6* mutants (**E**). (**D**′,**E**′) Traces of MC cells were made to enable easier analysis. (**F**) A cell was divided into five regions relative to the MC midline and the position of MTOC was assigned to each region (left schematic). MTOC location should be skewed equally to the left and right positions; however, mutants displayed more randomization with greater number of MTOC in the bottom and center locations. A total of 4 WT and 4 *dlx5i6^−/−^* larvae were analyzed, representing 55 cells for each genotype. * *p* < 0.05.

**Table 1 biomolecules-13-01347-t001:** Sequences to generate gRNA.

Target	gRNA 1 (5′-3′)	gRNA 2 (5′-3′)
*dlx1a*	GGACCAATCAGAGAGCACCT	GGTCCCCTGAACTGGGCCAT
*dlx2a*	GGCAATGATCAACGTGGCAT	GGAAACGCTTTCGGCCCCTA
*dlx5a*	GGCTATTACAGCCCTGCCGG	GGCTCTCGATATAATGGCAT
*dlx6a*	GGGAGCCGGAATGGAGACAG and GGCAGGTACGGACTGTGGTG	GGCGTGTCAATGGTCGACAA
*inter56*	GGCGTGTCAATGGTCGACAA	GGCTCTCGATATAATGGCAT
*dlx5i6*	GGGAGCCGGAATGGAGACAG	GGCTATTACAGCCCTGCCGG

**Table 2 biomolecules-13-01347-t002:** Primers for genotyping *dlx* mutants.

Target	Forward Sequence (5′-3′)	Reverse Sequence	Product Size
*dlx1a* WT	CCTATGCGTCTCGGTCCATT	CGTAGTGCGTCGACAAAAGC	740 bp
*dlx1a mutant*		GTGTTTCTTCTCCGGTGCGA	320–350 bp
*dlx2a* WT	GCATGAAACACATGGAAACCGA	AGTCTCCACCTCACCCCTCA	629 bp
*dlx2a mutant*	TTGCTGACGCGACATTGC	CAAGTCCCAGGAATCGCCG	130–160 bp
*dlx5a* WT	ATCAAGAACCAGGCGCATCT	GAGCCCACACAGAGATAGCA	498 bp
*dlx5a mutant*		CCCTTTTCAGCTCTCCACGAT	290–310 bp
*dlx6a WT*	GGAGGCTCAAGACTCGTCAA	TTGCTGACATACGACGGGG	429 bp
*dlx6a mutant*		AAAACGATTCGCTTCCTGTC	200–250 bp
*inter56 WT*	AGCTACATGCCCGGCTATTC	GGACAAGATGCGGCTCTATTC	796 bp
*inter56 mutant*		ATAGCCCCCAACCTACAAGC	350–400 bp
*dlx5i6 WT*	GGAGGCTCAAGACTCGTCAA	TTGCTGACATACGACGGGG	429 bp
*dlx5i6 mutant*		CCCTTTTCAGCTCTCCACGAT	200–230 bp

**Table 3 biomolecules-13-01347-t003:** Oligo sequences used to generate RNA probes.

Target	Forward Sequence (5′-3′)	Reverse Sequence (5′-3′)
*foxd3*	AATTAACCCTCACTAAAGGCTCAGTGGA	TAATACGACTCACTATAGCCATTTCGAT
	ATCTGCGAGT	ACCGCTGCTG
*col11a2*	AATTAACCCTCACTAAAGAGACCTCGTTT	TAATACGACTCACTATAGACGGGCTCC
	GTGCTCCTC	AAAAACAGTGA
*wls*	AATTAACCCTCACTAAAGCGGATGGAGC	TAATACGACTCACTATAGCATTGCGCA
	TTCGATCACC	GGCAGTAATCC
*wnt5b*	AATTAACCCTCACTAAAGGTGACCCTCAT	TAATACGACTCACTATAGCATTGCGCA
	CGTCTGCAA	GGCAGTAATCC

**Table 4 biomolecules-13-01347-t004:** Oligo sequences used for RT-qPCR.

Target	Forward Sequence (5′-3′)	Reverse Sequence (5′-3′)	Amplicon Size
*sox9a*	GCGTCCAGCATGGGAGAAGT	GCCTCTCGTTTCAGATCCGC	125 bp
*col11a1* [33]	GATATTCGGAAGAAGCGGAGG	CGCAAAACATCTACTGGATCTG	101 bp
*wnt5b*	TATTGGAGAGGGGGCGAAGA	ATATGCATGACACGGCCGAA	114 bp
*ror2* [34]	GAGGATTACAACTGGAGCTCAT	AGCCTCAATCTCTTCAGACTCG	106 bp
*frzd7a*	GACAAATTCAGCGACGACGG	CCCGCTGAGAGAAACCATGT	149 bp

## Data Availability

The original contributions presented in the study are included in the article/Appendix A. Further inquiries can be directed to the corresponding author.

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
