# Peer review of "Loss of dlx5a/dlx6a Locus Alters Non-Canonical Wnt Signaling and Meckel’s Cartilage Morphology"

_biomolecules, 2023, doi:10.3390/biom13091347_

Round 1
Author Response
Thank you very much for your comments. Each point will be addressed in order below:
“Regarding Figure 1 references in text…”
All figure references throughout the manuscript are now addressing the correct figures. Figure captions were also modified to correctly address figure components. For Figure 1 specifically, an additional panel was included to highlight the different anatomical landmarks used and how they were measured. Some font sizes were enlarged as were the graphs however they were still limited by the number of panels required for this Figure.
“In Figure 3, since capital letters were used…”
Figure changed to address this comment.
“In Figure 4 panels A and B…”
Arrows are now present in those figure panels to indicate apoptotic cells.
“When the authors apply the in situ hybridization technique…”
For sox9a, the staining in 3dpf mutant larvae was more intense and the entirety of cartilage structures in the MC, CH and ceratobranchials were stained compared to the lighter staining in WT. We referred to the additional area of staining as “expanded” however the authors agree the wording is vague and addressed the relevant text accordingly (lines 325-329). The description of staining changes for col11a2 was also altered to better reflect the provided images (lines 330-333). While in situ is indeed a qualitative assay, relative differences in staining intensity could indicate differences in expression levels between WT and mutant since they were processed at the same time and both biological groups were stained for the same amount of time. The qPCR was done to provide additional information about gene expression and while no statistical significance was demonstrated, a trend could still be informative. The entire head was processed for RNA so the RNA from the additional tissues present could obscure expression differences of craniofacial tissues.
“… the non-canonical wnt signaling is misexpressed…”
While the qPCR only showed a trend of misexpression, the authors decided this was useful information to provide, especially because the quantitative trend corresponded with in situ results. As with the above point, it is possible only a trend was observed due to additional tissues obscuring craniofacial expression of Wnt signaling components and this is addressed in the discussion (lines 468-472). Finally, since expression differences did not reach significance, we performed further assays to test whether these “trends” led to dysfunction of non-canonical Wnt signaling. Downstream targets of non-canonical Wnt signaling pathways, such as JNK/C-jun or NFAT expression/function could be affected; this point is now addressed in the discussion (lines 482-487). It is also possible that other factors are involved alternatively or due directly to misexpressed wnt signaling, such as fgf3 which is regulated by Wnt5b activity. These additional factors are now mentioned in the discussion section and may be the focus of future studies (lines 487-489).
“In line 391..”
The reference to supplementary figure 2 is removed and replaced with a reference to a previously published paper that does show loss of dlx5a result in no expression of dlx6a.
“Figure 7A has poor quality and is difficult to observe Meckel’s cartilage..”
The image is slightly enlarged for better visualization. However, at the timepoint the image is taken, the Meckel’s cartilage was still developing and will not have defined chondrocyte stacking, particularly in the middle section. Previously published examples of MTOC localization in the Meckel’s cartilage also display this incomplete definition of cartilage structure at 3dpf (see for example Ling et al., 2017- Fig. 3I-M).
“… include the primers used to genotype…”
A list of primers is now included under Table 2 (page 3).
“… acridine orange staining is not possible to understand…”
This is now addressed in the “Imaging” section at lines 199-200.
“In line 847… performed time lapse imaging…”
This line is now removed.
“All sections of the paper… is repeated.”
The repeated sections were not part of the submitted manuscript. They appeared when our manuscript was converted to the MDPI template by the Journal. The revised manuscript is now in the correct format.
All minor comments were addressed in the appropriate sections.

Reviewer 2 Report
The article entitled "Loss of dlx5a/dlx6a locus alters non-canonical Wnt signaling and Meckel’s cartilage morphology ' by Yu et al explains important information about the dlx gene function in facial cartilage morphogenesis. I enjoyed reading this manuscript. However, this needs major revisions with regards to the arrangement of the content. There are two methods section and two results sections. Please check and revised accordingly.
The figures can be improved with labeling.

Minor editing of English language required
Author Response
Thank you for reviewing our manuscript. Below you will find our responses to your comments in order.
“Please explain what are the land marks…” and “Label the structures…”
An additional panel was added into Figure 1 (Fig. 1L) that labels the structures measured and how they were measured.
“Can you comment about this rescue effect?”
We did not do any analysis on craniofacial structures in adults. They appear to be normal just by looking at their head as they swim and they can eat normally. Whether the abnormal morphology of the Meckel’s cartilage persists or if additional modifications developed to compensate for the change is unknown.
“This information can be moved…”
The repeated and misplaced sections were not part of the submitted manuscript. They appeared when our manuscript was converted to the MDPI template by the Journal. The revised manuscript is now in the correct format.

Reviewer 3 Report
This manuscript primarily shows that, when dlx5a/dlx6a is deleted, abnormal craniofacial development. That’s an interesting finding that demonstrated in vivo function of dlx5a and dlx6a. This appears to be a novel observation. This manuscript results and discussion are well done, but I feel that some data need to be corrected.
1) Descriptions from Line 447 to Line 723 are duplicated. Please confirm.
2) There is an impression that the intensity of the photographs is remarkably difference between Fig. 1A-G and Fig. 2A-G. Please consider changing to photographs with the same exposure intensity
3) For all of image data, please consider inserting a scale into every individual image.
4) It is not appropriate to write the n-value in the image, for example Fig.4. Please write it in each figure legends.
Author Response
Thank you for taking the time to read our manuscript. Below, we have addressed your comments in order.
“Descriptions from Line 447-723…”
The repeated sections were not part of the submitted manuscript. They appeared when our manuscript was converted to the MDPI template by the Journal. The revised manuscript is now in the correct format.
“… impression that the intensity of photographs is remarkably different between Fig. 1A-G and Fig. 2A-G…”
The exposure was set to the same numbers during imaging on the program and manual changing of light intensity was kept to the same as much as possible. Since the staining of each mutant and their WT was done on different days and images taken on different days, the staining of structures would be different. Images were chosen based on how representative they were of all individuals imaged across at least 3 trials and of how well the structures were stained. We tried to choose images with similar intensity of stained structures.
The intensity of staining between Figure 1 and Figure 2 structures would be different since images in Figure 2 are of older fish.
“… inserting a scale into every individual image.”
The scale bar in Fig. 1A and Fig. 2A was made thicker to be more easily visible. A scale bar was added to Fig. 1A’ and Fig. 2A’ as well to indicate the images taken in the lateral orientation are of the same magnification and zoom as samples in the ventral position. However, the authors prefer not to include scale bars in every individual image as we feel it would distract from the point of each image without offering new information.
“.. n-values in the image…”
This comment is addressed. All n-values are removed from the figures and found in the figure captions instead.

Round 2
Reviewer 1 Report
After the major revision and the answers provided by the authors, I think the document is ready for publication.
Best Regards
Reviewer 2 Report
None